# Bayesian Optimization with Unknown Search Space

**Huong Ha, Santu Rana, Sunil Gupta, Thanh Nguyen, Hung Tran-The, Svetha Venkatesh**
Applied Artificial Intelligence Institute ($A^2I^2$)
Deakin University, Geelong, Australia
{huong.ha, santu.rana, sunil.gupta, thanhnt, hung.tranthe, svetha.venkatesh}@deakin.edu.au

## Abstract

Applying Bayesian optimization in problems wherein the search space is unknown is challenging. To address this problem, we propose a systematic volume expansion strategy for the Bayesian optimization. We devise a strategy to guarantee that in iterative expansions of the search space, our method can find a point whose function value within $\epsilon$ of the objective function maximum. Without the need to specify any parameters, our algorithm automatically triggers a minimal expansion required iteratively. We derive analytic expressions for when to trigger the expansion and by how much to expand. We also provide theoretical analysis to show that our method achieves $\epsilon$-*accuracy* after a finite number of iterations. We demonstrate our method on both benchmark test functions and machine learning hyper-parameter tuning tasks and demonstrate that our method outperforms baselines.

## 1 Introduction

Choosing where to search matters. A time-tested path in the quest for new products or processes is through experimental optimization. Bayesian optimization offers a sample efficient strategy for experimental design by optimizing expensive black-box functions [9–11]. But one problem is that users need to specify a bounded region to restrict the search of the objective function extrema. When tackling a completely new problem, users do not have prior knowledge, hence there is no guarantee that an arbitrarily defined search space contains the global optimum. Thus application of the Bayesian optimization framework when the search region is unknown remains an open challenge [16].

One approach is to use a regularized acquisition function such that its maximum can never be at infinity - hence no search space needs to be declared and an unconstrained optimizer can be used [16]. Other approaches use volume expansion, i.e. starting from the user-defined region, the search space is expanded during the optimization. The simplest strategy is to repeatedly double the volume of the search space every several iterations [16]. Nguyen et al suggest a volume expansion strategy based on the evaluation budget [12]. All these methods require users to specify critical parameters - as example, regularization parameters [16], or growth rate, expansion frequency (volume doubling) [16] or budget [12]. These parameters are difficult to specify in practice. Additionally, [12] is computationally expensive and the user-defined search space needs to be close to the global optimum.

In this paper, we propose a systematic volume expansion strategy for the Bayesian optimization framework wherein the search space is unknown. Without any prior knowledge about the objective function argmax or strict assumptions on the behavior of the objective function, it is impossible to guarantee the global convergence when the search space is continuously expanded. To circumvent this problem, we consider the setting where we achieve the *global $\epsilon$-accuracy* condition, that is, we aim to find a point whose function value is within $\epsilon$ of the objective function global maximum.

Our volume expansion strategy is based on two guiding principles: 1) The algorithm can reach a point whose function value is within $\epsilon$ of the objective function maximum in one expansion, and, 2) the search space should be minimally expanded so that the algorithm does not spend unnecessary

evaluations near the search space boundary. As the objective function is unknown, it is not possible to compute this ideal expansion region. Using the GP-UCB acquisition function as a surrogate, this region is computed as one that contains at least one point whose acquisition function value is within $\epsilon$ of the acquisition function maximum. However, by using a surrogate to approximate the objective function, there is no guarantee that we can achieve the *global $\epsilon$-accuracy* within one expansion. Hence multiple expansions are required, and a new expansion is triggered when the *local $\epsilon$-accuracy* is satisfied, i.e. when the algorithm can find a point whose function value is within $\epsilon$ of the objective function maximum in the current search space. Analytical expressions for the size of the new expansion space and when to trigger the expansion are derived. The guarantees for the $\epsilon$-accuracy condition, however, now lapses in the expanded region, and so we adjust the acquisition function appropriately to maintain the guarantee. Finally, we provide theoretical analysis to show that our proposed method achieves the *global $\epsilon$-accuracy* condition after a finite number of iterations.

We demonstrate our algorithm on five synthetic benchmark functions and three real hyperparameter tuning tasks for common machine learning models: linear regression with elastic net, multilayer perceptron and convolutional neural network. Our experimental results show that our method achieves better function values with fewer samples compared to state-of-the-art approaches. In summary, our contributions are:

- Formalising the analysis for Bayesian optimization framework in an unknown search space setting, and introducing *$\epsilon$-accuracy* as a way to track the algorithmic performance;

- Providing analytic expressions for how far to expand the search space and when to expand the search space to achieve *global $\epsilon$-accuracy*;

- Deriving theoretical *global $\epsilon$-accuracy* convergence; and,

- Demonstrating our algorithm on both synthetic and real-world problems and comparing it against state-of-the-art methods.

Our method differs from previous works in that 1) our method does not require any algorithmic parameters, automatically adjusting both when to trigger the expansion and by how much to expand, and, 2) our approach is the only one to guarantee the *global $\epsilon$-accuracy* condition. This is because we guarantee the *local $\epsilon$-accuracy* condition in each search space, thus eventually the *global $\epsilon$-accuracy* is achieved. Without this local guarantee, the suggested solution cannot be guaranteed to reach *global $\epsilon$-accuracy*. The regularization [16] and the filtering method [12] require the global optimum to be within a bound constructed by either the user specified regularizer or the budget. The volume doubling method [16] can continue to expand the search space to infinity, however, the *local $\epsilon$-accuracy* condition is not guaranteed in each search space.

The paper is organized as follows. Section 2 gives an overview of Bayesian optimization and discusses some of the related work. Section 3 describes the problem setup. Section 4 proposes our new expansion strategy for the Bayesian optimization framework when the search space is unknown. A theoretical analysis for our proposed method is presented in Section 5. In Section 6, we demonstrate the effectiveness of our algorithm by numerical experiments. Finally, Section 7 concludes the paper.

## 2 Background and Related Work

### 2.1 Background

Bayesian optimization is a powerful optimization method to find the global optimum of an unknown objective function $f(x)$ by sequential queries [9–11, 17, 18]. First, at time $t$, a surrogate model is used to approximate the behaviour of $f(x)$ using all the current observed data $\mathcal{D}_{t-1} = \{(x_i, y_i)\}_{i=1}^{n}$, $y_i = f(x_i) + \xi_i$, where $\xi_i \sim \mathcal{N}(0, \sigma^2)$ is the noise. Second, an acquisition function is constructed from the surrogate model that suggests the next point $x_{itr}$ to be evaluated. The objective function is then evaluated at $x_{itr}$ and the new data point $(x_{itr}, y_{itr})$ is added to $\mathcal{D}_{t-1}$. These steps are conducted in an iterative manner to get the best estimate of the global optimum.

The most common choice for the surrogate model used in Bayesian optimization is the Gaussian Process (GP) [14]. Assume the function $f$ follows a GP with mean function $m_0(x)$ and covariance function $k(x, x')$, the posterior distribution of $f$ given the observed data $\mathcal{D}_{t-1} = \{(x_i, y_i)\}_{i=1}^{n}$ is a

GP with the following posterior mean and variance,

$$\mu_{t-1}(x) = m_0(x) + \mathbf{k}_{|\mathcal{D}_{t-1}|}(x)^T (\mathbf{K}_{|\mathcal{D}_{t-1}|} + \sigma^2 \mathbf{I}_{|\mathcal{D}_{t-1}|})^{-1} \mathbf{y}_{|\mathcal{D}_{t-1}|},$$
$$\sigma_{t-1}^2(x) = k(x,x) - \mathbf{k}_{|\mathcal{D}_{t-1}|}(x)^T (\mathbf{K}_{|\mathcal{D}_{t-1}|} + \sigma^2 \mathbf{I}_{|\mathcal{D}_{t-1}|})^{-1} \mathbf{k}_{|\mathcal{D}_{t-1}|}(x),$$

(1)

where $\mathbf{y}_{|\mathcal{D}_{t-1}|} = [y_1, \ldots, y_{|\mathcal{D}_{t-1}|}]^T$, $\mathbf{k}_{|\mathcal{D}_{t-1}|}(x) = [k(x, x_i)]_{i=1}^{|\mathcal{D}_{t-1}|}$, $\mathbf{K}_{|\mathcal{D}_{t-1}|} = [k(x_i, x_j)]_{i,j}$, $\mathbf{I}_{|\mathcal{D}_{t-1}|}$ is the $|\mathcal{D}_{t-1}| \times |\mathcal{D}_{t-1}|$ identity matrix and $|\mathcal{D}_{t-1}|$ denotes the cardinality of $\mathcal{D}_{t-1}$. To aid readability, in the sequel we remove the notation that shows the dependence of $\mathbf{k}, \mathbf{K}, \mathbf{I}, \mathbf{y}$ on $|\mathcal{D}_{t-1}|$.

There are many existing acquisition functions [6, 7, 10, 11, 20] and in this paper, we focus only on the GP-UCB acquisition function [1, 2, 5, 19]. The GP-UCB acquisition function is defined as,

$$\alpha_{UCB}(x; \mathcal{D}_{t-1}) = \mu_{t-1}(x) + \sqrt{\beta_t} \sigma_{t-1}(x),$$

(2)

where $\mu_{t-1}(x), \sigma_{t-1}(x)$ are the posterior mean and standard deviation of the GP given observed data $\mathcal{D}_{t-1}$ and $\beta_t \geq 0$ is an appropriate parameter that balances the exploration and exploitation. Given a search domain, $\{\beta_t\}$ can be chosen as in [19] to ensure global convergence in this domain.

## 2.2 Related Work

All the work related to the problem of Bayesian optimization with unknown search space have been described in Section 1. There is the work in [3] introduces the term $\epsilon$-*accuracy*. However, their purpose is to unify the Bayesian optimization and the Level-set estimation framework.

## 3 Problem Setup

We wish to find the global argmax $x_{max}$ of an unknown objective function $f : \mathbb{R}^d \mapsto \mathbb{R}$, whose argmax is at a finite location, i.e.

$$x_{max} = \mathrm{argmax}_{x \in \mathcal{S}_*} f(x),$$

(3)

where $\mathcal{S}_*$ is a finite region that contains the argmax of the function $f(x)$. In practice, the region $\mathcal{S}_*$ is not known in advance, so users need to identify a search domain $\mathcal{S}_{user}$ which is likely to contain the argmax of $f(x)$. This search domain can be set arbitrarily or based on limited prior knowledge. Thus there is no guarantee that $\mathcal{S}_{user}$ contains the global optimum of the objective function. In the trivial cases when the search space $\mathcal{S}_*$ is known or when $\mathcal{S}_* \subset \mathcal{S}_{user}$, the global convergence can be guaranteed through classical analysis [4, 19]. Here, we consider the general case when $\mathcal{S}_*$ may or may not be a subset of $\mathcal{S}_{user}$. Without any prior knowledge about $\mathcal{S}_*$ or strict assumptions on the behavior of the objective function, it is impossible to guarantee the global convergence. Therefore, in this work, instead of solving Eq. (3), we consider the setting where we achieve the *global $\epsilon$-accuracy* condition. That is, for a small positive value $\epsilon$, we find a solution $x_\epsilon$ which satisfies,

$$f(x_{max}) - f(x_\epsilon) \leq \epsilon.$$

(4)

## 4 Proposed Approach

We make some mild assumptions to develop our main results.

**Assumption 4.1** *The prior mean function $m_0(x) = 0$.*

This is done by subtracting the mean from all observations and is common practice.

**Assumption 4.2** *The kernel $k(x, x')$ satisfies, (1) when $\|x - x'\|_2 \to +\infty$, $k(x, x') \to 0$; (2) $k(x, x') \leq 1 \ \forall (x, x')$ ; (3) $k(x, x) = \theta^2$, where $\theta \geq 0$ is the scale factor of the kernel function.*

Various kernels satisfy Assumption 4.2, e.g. the Matérn kernel, the Square Exponential kernel. As the function can always be re-scaled, condition 2 is met without loss of generality [15, 19].

**Defining $g_k(\gamma)$:** With these types of kernels, for all small positive $\gamma$, there always exists $g_k(\gamma) > 0$,

$$\forall x, x' : \|x - x'\|_2 \geq g_k(\gamma), \quad k(x, x') \leq \gamma.$$

(5)

The value of $g_k(\gamma)$ can be computed from $\gamma$ and the kernel covariance function $k(x, x')$ i.e. for Squared Exponential kernel $k_{SE}(x, x') = \theta^2 \exp(-\|x - x'\|_2^2/(2l^2))$, $g_k(\gamma)$ will be $\sqrt{2l^2 \log(\theta^2/\gamma)}$.

**Assumption 4.3** *The kernel $k(x, x')$ is known in advance or can be learned from the observations.*

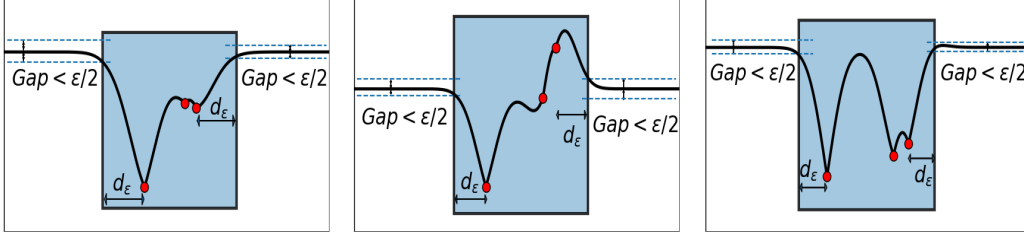

Figure 1: Expanded region (blue), when the GP-UCB acquisition function argmax is at (1) infinity ; or (2) at a finite location and its function value is larger or equal $\sqrt{\beta_t}\theta + \epsilon/2$; or (3) at a finite location and its function value is smaller than $\sqrt{\beta_t}\theta + \epsilon/2$.

## 4.1 Proposed Expansion Strategy

The ideal expansion strategy should satisfy two characteristics: 1) The algorithm can reach the *global $\epsilon$-accuracy* condition in one expansion, and, 2) the search space should be minimally expanded so that the algorithm does not spend unnecessary evaluations near the search space boundary. Since we have a black-box objective function, it is not possible to compute the ideal expansion space $\mathcal{S}_{ideal}$ directly. Let the exploration-exploitation parameters $\{\beta_t\}$ be chosen to ensure the objective function is upper bounded by the GP-UCB acquisition function with high probability. Then we can estimate $\mathcal{S}_{ideal}$ by a region $\mathcal{S}$ as a minimal region that contains at least one point whose acquisition function value is within $\epsilon$ from the acquisition function maximum, i.e. $\exists x_u \in \mathcal{S}$ : $|\alpha_{UCB}(x_u; \mathcal{D}_{t-1}) - \max_{x \in \mathbb{R}^d} \alpha_{UCB}(x; \mathcal{D}_{t-1})| \leq \epsilon$. Due to the approximation, there is no guarantee we can achieve the *global $\epsilon$-accuracy* in one expansion. Thus we need multiple expansions sequential. A new expansion is triggered when the *local $\epsilon$-accuracy* is satisfied in the previous expansion. In the following, we first derive the value of the GP-UCB acquisition function when $x \to \infty$ (Proposition 4.1), and then use this value to derive analytical expressions for the size of the expansion space $\mathcal{S}$ (Theorem 4.1) and when to trigger a new expansion.

**Proposition 4.1** *When $x \to \infty$, the GP-UCB acquisition function $\alpha_{UCB}(x; \mathcal{D}_{t-1}) \to \sqrt{\beta_t}\theta$, where $\beta_t$ is the exploration-exploitation parameter of the GP-UCB acquisition function and $\theta$ is the scale factor of the kernel function $k(x, x')$.*

**Derivation of the expansion search space**   Our idea is to choose the region $\mathcal{S}$ such that $\mathcal{S} = \mathbb{R}^d \setminus \mathcal{A}$, where 1) $\mathcal{A}$ contains all the points $x$ that are far from all the current observations, and, 2) $\mathcal{A} := \{x \in \mathbb{R}^d : |\alpha_{UCB}(x; \mathcal{D}_{t-1}) - \sqrt{\beta_t}\theta| < \epsilon/2\}$. Here, we will show that with this choice of $\mathcal{S}$, there exists at least one point in $\mathcal{S}$ whose acquisition function value is within $\epsilon$ from the acquisition function maximum, given $\epsilon < |\sqrt{\beta_t}\theta - \min_{x \in \mathbb{R}^d}(\alpha_{UCB}(x; \mathcal{D}_{t-1}))|$. We consider three cases that can happen to the GP-UCB acquisition function (See Figure 1):

- **Case 1**: The argmax of the GP-UCB acquisition function is at infinity. This means that the GP-UCB acquisition function maximum is equal to $\sqrt{\beta_t}\theta$. As the GP-UCB acquisition function is continuous and $\epsilon < |\sqrt{\beta_t}\theta - \min_{x \in \mathbb{R}^d}(\alpha_{UCB}(x; \mathcal{D}_{t-1}))|$, hence, there exists a point $x_u$ such that $\alpha_{UCB}(x_u) = \sqrt{\beta_t}\theta - \epsilon/2$. By the definition of $\mathcal{S}$, it is straightforward that $x_u$ belongs to $\mathcal{S}$, thus proving that there exists a point in $\mathcal{S}$ whose GP-UCB acquisition function value is within $\epsilon$ from the maximum of the acquisition function.

- **Case 2**: The argmax of the GP-UCB acquisition function $x'_{max}$ is at a finite location and its acquisition function value is larger or equal $\sqrt{\beta_t}\theta + \epsilon/2$. It is straightforward to see that the argmax $x'_{max}$ belongs to the region $\mathcal{S}$ and this is the point that satisfies $|\alpha_{UCB}(x'_{max}; \mathcal{D}_{t-1}) - \max_{x \in \mathbb{R}^d} \alpha_{UCB}(x; \mathcal{D}_{t-1})| \leq \epsilon$.

- **Case 3**: The GP-UCB acquisition function argmax is at a finite location and the acquisition function maximum is smaller than $\sqrt{\beta_t}\theta + \epsilon/2$. As the GP-UCB acquisition function is continuous and $\epsilon < |\sqrt{\beta_t}\theta - \min_{x \in \mathbb{R}^d}(\alpha_{UCB}(x; \mathcal{D}_{t-1}))|$, there exists a point $x_u \in \mathcal{S}$ : $\alpha_{UCB}(x_u; \mathcal{D}_{t-1}) = \sqrt{\beta_t}\theta - \epsilon/2$. As $\max_{x \in \mathbb{R}^d} \alpha_{UCB}(x; \mathcal{D}_{t-1}) < \sqrt{\beta_t}\theta + \epsilon/2$, it follows directly that $|\alpha_{UCB}(x_u; \mathcal{D}_{t-1}) - \max_{x \in \mathbb{R}^d} \alpha_{UCB}(x; \mathcal{D}_{t-1})| \leq \epsilon$.

Theorem 4.1 now formally derives an analytical expression for one way to define region $\mathcal{S}$.

**Algorithm 1** Bayesian optimization with unknown search space (GPUCB-UBO)

---

1: **Input:** Gaussian Process (GP) $\mathcal{M}$, acquisition functions $\alpha_{UCB}, \alpha_{LCB}$, initial observations $\mathcal{D}_{init}$, initial search space $S_{user}$, function $f$, positive small threshold $\epsilon$, evaluation budget $T$.
2: **Output:** Point $x_\epsilon : \max f(x) - f(x_\epsilon) \leq \epsilon$.
3: Initialize $\mathcal{D}_0 = \mathcal{D}_{init}$, $\mathcal{S} = \mathcal{S}_{user}$, $\beta_1$, $t_k = 0$. Update the GP using $\mathcal{D}_0$.
4: **for** $t = 1, 2, \ldots, T$ **do**
5:     Set $t_{local} = t - t_k$
6:     Compute $x_m = \text{argmax}_{x \in \mathcal{S}}\, \alpha_{UCB}(x; \mathcal{D}_{t-1})$
7:     Set $x_t = x_m$, $y_t = f(x_t)$. Update $\mathcal{D}_t = \mathcal{D}_{t-1} \cup (x_t, y_t)$.
8:     */* Compute the expansion trigger, the regret upper bound */*
9:     Compute $r_b = \alpha_{UCB}(x_t; \mathcal{D}_{t-1}) - \max_{x \in \mathcal{D}_t} \alpha_{LCB}(x; \mathcal{D}_{t-1}) + 1/t_{local}^2$
10:    */* If expansion triggered, expand the search space */*
11:    **if** $(r_b <= \epsilon) \,|\, (t == 1)$ **then**
12:      Compute the new search space $\mathcal{S}$ as defined in Theorem 4.1
13:      Set $t_k = t_k + t_{local}$
14:    **end if**
15:    */* Adjust the $\beta_t$ based on the search space */*
16:    Compute $\beta_t$ following Theorem 5.1
17:    Update the GP using $\mathcal{D}_t$.
18: **end for**

---

**Theorem 4.1** *Consider the GP-UCB acquisition function $\alpha_{UCB}(x; \mathcal{D}_{t-1})$. Let us define the region $\mathcal{S} = \bigcup_{i=1}^{|\mathcal{D}_{t-1}|} \mathcal{S}_i$, $\mathcal{S}_i = \{x : \|x - x_i\|_2 \leq d_\epsilon\}$, $x_i \in \mathcal{D}_{t-1}$, $|\mathcal{D}_{t-1}|$ is the cardinality of $\mathcal{D}_{t-1}$, $d_\epsilon = g_k(min(\sqrt{(\sqrt{\beta_t}\theta\epsilon/2 - \epsilon^2/16)/(|\mathcal{D}_{t-1}|\lambda_{max})}/\sqrt{\beta_t},\ 0.25\epsilon/max(\sum_{z_j \leq 0} -z_j, \sum_{z_j \geq 0} z_j)))$ with $g_k(.)$ as in Eq. (5), $\lambda_{max}$ be the largest singular value of $(\boldsymbol{K} + \sigma^2\boldsymbol{I})^{-1}$, and $z_j$ be the $j^{th}$ element of $(\boldsymbol{K} + \sigma^2\boldsymbol{I})^{-1}\boldsymbol{y}$. Given $\epsilon < |\sqrt{\beta_t}\theta - \min_{x \in \mathbb{R}^d}(\alpha_{UCB}(x; \mathcal{D}_{t-1}))|$, then there exists at least one point in $\mathcal{S}$ whose acquisition function value is within $\epsilon$ from the acquisition function maximum, i.e. $\exists x_u \in \mathcal{S} : |\alpha_{UCB}(x_u; \mathcal{D}_{t-1}) - \max_{x \in \mathbb{R}^d} \alpha_{UCB}(x; \mathcal{D}_{t-1})| \leq \epsilon$.*

**Acquisition function adaption**    Let us denote $\mathcal{S}_k$ as the $k^{th}$ expansion search space ($k \geq 1$). In each $\mathcal{S}_k$, the parameter $\{\beta_t\}$ of the GP-UCB acquisition function needs to be valid to ensure the algorithm achieves the *local $\epsilon$-accuracy* condition. Hence, a new $\{\beta_t\}$ is adjusted after each expansion. Details on how to compute the new $\{\beta_t\}$ are in Theorem 5.1.

**Triggering the next expansion**    To guarantee the *global $\epsilon$-accuracy* condition, in each search space $\mathcal{S}_k$, we aim to find an iteration $T_k$ which satisfies $r_{\mathcal{S}_k}(T_k) = (\max_{x \in \mathcal{S}_k} f(x) - \max_{x_i \in \mathcal{D}_{T_k}} f(x_i)) \leq \epsilon$ before the next expansion. As we do not have $\max_{x \in \mathcal{S}_k} f(x)$ and $\{f(x_i)\}$, we bound $r_{\mathcal{S}_k}(t)$ by $r_{b,\mathcal{S}_k}(t) = \max_{x \in \mathcal{S}_k} \alpha_{UCB}(x; \mathcal{D}_{t-1}) + 1/t^2 - \max_{x \in \mathcal{D}_t} \alpha_{LCB}(x; \mathcal{D}_{t-1})$, where $\alpha_{LCB}(x; \mathcal{D}_{t-1}) = \mu_{t-1}(x) - \sqrt{\beta_t}\sigma_{t-1}(x)$. The next expansion is triggered when $r_{b,\mathcal{S}_k}(t)$ reaches $\epsilon$.

**Search space optimization**    The theoretical search space developed in Theorem 4.1 is the union of $|\mathcal{D}_{t-1}|$ balls. To suit optimizer input, this region is converted to an encompassing hypercube using,

$$\min_{x_i \in \mathcal{D}_{t-1}}(x_i^k) - d_\epsilon \leq x^k \leq \max_{x_i \in \mathcal{D}_{t-1}}(x_i^k) + d_\epsilon,\ k = \overline{1, d}. \tag{6}$$

Further refinement of the implementation is provided in the supplementary material.

**Algorithm 1** describes the proposed Bayesian optimization with unknown search space algorithm.

## 5   Theoretical Analysis

First, to ensure the validity of our algorithm, we prove that for a wide range of kernels, for any search space $\mathcal{S}_k$ and any positive $\epsilon$, with a proper choice of $\{\beta_t\}$, our trigger for expansion condition occurs with high probability. When this happens, the algorithm achieves the *local $\epsilon$-accuracy* condition.

**Proposition 5.1** *For any $d$-dimensional domain $\mathcal{S}_k$ with side length $r_k$, for the kernel classes: finite dimensional linear, Squared Exponential and Matérn, suppose the kernel $k(x, x')$ satisfies the following condition on the derivatives of GP sample paths $f$: $\exists a_k, b_k > 0$, $Pr\{\sup_{x \in \mathcal{S}_k} |\partial f / \partial x_j| >$*

$L\} \leq a_k \exp^{-(L/b_k)^2}, j = \overline{1,d}$. *Pick* $\delta \in (0,1)$, *and define* $\beta_t = 2\log(t^2 2\pi^2/(3\delta)) + 2d\log(t^2 db_k r_k \sqrt{\log(4da_k/\delta)})$, *then* $\forall \epsilon > 0$, *with probability larger than* $1-\delta$, *there* $\exists T_k : \forall t \geq T_k, \max_{x \in \mathcal{S}_k} \alpha_{UCB}(x; \mathcal{D}_{t-1}) - \max_{x \in \mathcal{D}_t} \alpha_{LCB}(x; \mathcal{D}_{t-1}) \leq \epsilon - 1/t^2$; *and* $\forall t$ *that satisfies the previous condition*, $\max_{x \in \mathcal{S}_k} f(x) - \max_{x \in \mathcal{D}_t} f(x) \leq \epsilon$.

Second, we prove that with a proper choice of $\{\beta_t\}$ and for a wide range class of kernels, after a finite number of iterations, our algorithm achieves the *global $\epsilon$-accuracy* condition with high probability.

**Theorem 5.1** *Denote* $\{\mathcal{S}_k\}$ *as the series of the expansion search space suggested by our algorithm* $(k \geq 1)$. *In each* $\mathcal{S}_k$, *let* $T_k$ *be the smallest number of iterations that satisfies our expansion triggered condition, i.e.* $r_{b,\mathcal{S}_k}(T_k) \leq \epsilon$. *Suppose the kernel* $k(x,x')$ *belong to the kernel classes listed in Proposition 5.1 and it satisfies the following condition on the derivatives of GP sample paths* $f$: $\exists a_k, b_k > 0$, $Pr\{\sup_{x \in \mathcal{S}_k} |\partial f/\partial x_j| > L\} \leq a_k \exp^{-(L/b_k)^2}, j = \overline{1,d}$. *Pick* $\delta \in (0,1)$, *and define*, $\beta_t = 2\log((t - \sum_{j \leq k-1} T_j)^2 2\pi^2/(3\delta)) + 2d\log((t - \sum_{j \leq k-1} T_j)^2 db_k r_k \sqrt{\log(4da_k/\delta)})$, $\sum_{j \leq k-1} T_j + 1 \leq t \leq \sum_{j \leq k} T_j, k = 1, 2, \dots$ *Then running the proposed algorithm with the above choice of* $\beta_t$ *for a sample* $f$ *of a GP with mean function zero and covariance function* $k(x,x')$, *after a finite number of iterations, we achieve global $\epsilon$-accuracy with at least* $1-\delta$ *probability, i.e.*

$$Pr\{f(x_{max}) - f(x_{suggest}) \leq \epsilon\} \geq 1 - \delta,$$

*where* $x_{suggest}$ *is the algorithm recommendation and* $x_{max}$ *is the objective function global argmax.*

**Discussion**    The difference between our method and previous works is that we guarantee the *local $\epsilon$-accuracy* condition in each search space, eventually achieving the *global $\epsilon$-accuracy*. Previous methods do not give this guarantee, and thus their final solution may not reach *global $\epsilon$-accuracy*.

# 6   Experimental Evaluation

We evaluate our method on five synthetic benchmark functions and three hyperparameter tuning tasks for common machine learning models. For problems with dimension $d$, the optimization evaluation budget is $10d$ (excluding initial $3d$ points following a latin hypercube sampling [8]). The experiments were repeated 30 and 20 times for the synthetic functions and machine learning hyperparameter tuning tasks respectively. For all algorithms, the Squared Exponential kernel is used, the GP models are fitted using the Maximum Likelihood method and the output observations $\{y_i\}$ are normalized $y_i \sim \mathcal{N}(0,1)$. As with previous GP-based algorithms that use confidence bounds [3, 19], our theoretical choice of $\{\beta_t\}$ in Theorem 5.1 is typically overly conservative. Hence, following the suggestion in [19], for any algorithms that use the GP-UCB acquisition, we scale $\beta_t$ down by a factor of 5. Finally, for the synthetic functions, $\epsilon$ is set at $0.05$ whist for the machine learning models, $\epsilon$ is set at $0.02$ as we require higher accuracy in these cases.

We compare our proposed method, **GPUCB-UBO**, with seven baselines: (1) **EI-Vanilla**: the vanilla Expected Improvement (EI); (2) **EI-Volx2**: the EI with the search space volume doubled every $3d$ iterations [16]; (3) **EI-H**: the Regularized EI with a hinge-quadratic prior mean where $\beta = 1$ and $R$ is the circumradius of the initial search space [16]; (4) **EI-Q**: the Regularized EI with a quadratic prior mean where the widths $w$ are set to those of the initial search space [16]; (5) **GPUCB-Vanilla**: the vanilla GP-UCB; (6) **GPUCB-Volx2**: the GP-UCB with the search space volume doubled every $3d$ iterations [16]; (7) **GPUCB-FBO**: the GP-UCB with the fitering expansion strategy in [12].

## 6.1   Visualization

We visualize our theoretical expansion search spaces derived in Theorem 4.1 on the Beale test function (Figure 2). We show the contour plots of the GP-UCB acquisition functions, and show both the observations (red stars) and the recommendation from the algorithm that correspond the acquisition function maximum (cyan stars). The initial user-defined search space (black rectangle) is expanded as per theoretical search spaces developed in Theorem 4.1 (yellow rectangles). Here we use Eq. (6) to plot the expansion search spaces, however, the spaces developed in Theorem 4.1 are tighter. The figure illustrates that when the argmax of the objective function is outside of the user-defined search space, with our search space expansion strategy, this argmax can be located within a finite number of expansions.

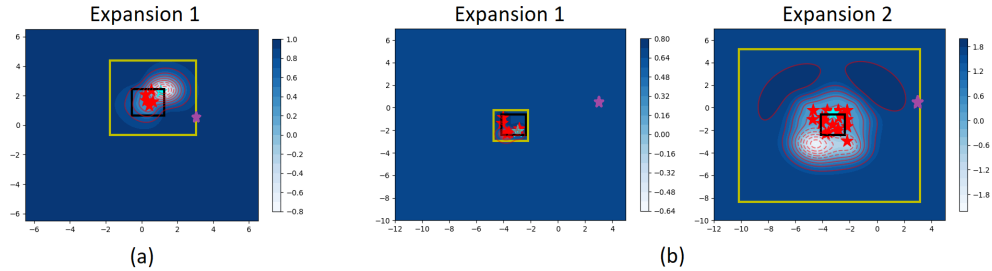

| | | |
|---|---|---|
| Expansion 1 | Expansion 1 | Expansion 2 |
| (a) | | (b) |

Figure 2: Expansion search spaces using Theorem 4.1 for Beale function in two cases when the *global ϵ-accuracy* is achieved within (a) one expansion; or (b) two expansions. The black rectangle is the user-defined search space and the yellow rectangles are the theoretical expansion search spaces. The contour plots of the acquisition function are also displayed with observations (red stars) and the recommendation at that iteration (cyan star). Global optimum of Beale function is the magenta star.

## 6.2 Synthetic Benchmarks

We compare our method with seven baselines on five benchmark test functions: Beale, Eggholder, Levy 3, Hartman 3 and Hartman 6. We use the same experiment setup as in [16]. The length of the initial user-defined search space is set to be $20\%$ of the length of the function domain - e.g. if the function domain is the unit hypercube $[0, 1]^d$, then the initial search space has side length of $0.2$. The center of this initial search space is placed randomly in the domain of the objective function.

For each test function and algorithm, we run the experiment 30 times, and each time the initial search space will be placed differently. We plot the mean and the standard error of the best found values $\max_{i=\overline{1,n}} f(x_i)$ of each test function. Figure 3 shows that for most test functions, our method GPUCB-UBO achieves both better function values and in less iterations than other methods. For most test functions, our method is better than other six state-of-the-art approaches (except GPUCB-FBO) by a high margin. Compared with GPUCB-FBO, our method is better on the test functions Hartman3 and Hartman6 while performing similar on other three test functions. Note that the computation time of GPUCB-FBO is 2-3 times slower than our method and other approaches (see Table 1) because it needs an extra step to numerically solve several optimization problems to construct the new search space. Since we derive the expansion search spaces analytically, our method, in contrast, can optimize the acquisition function within these spaces without any additional computation.

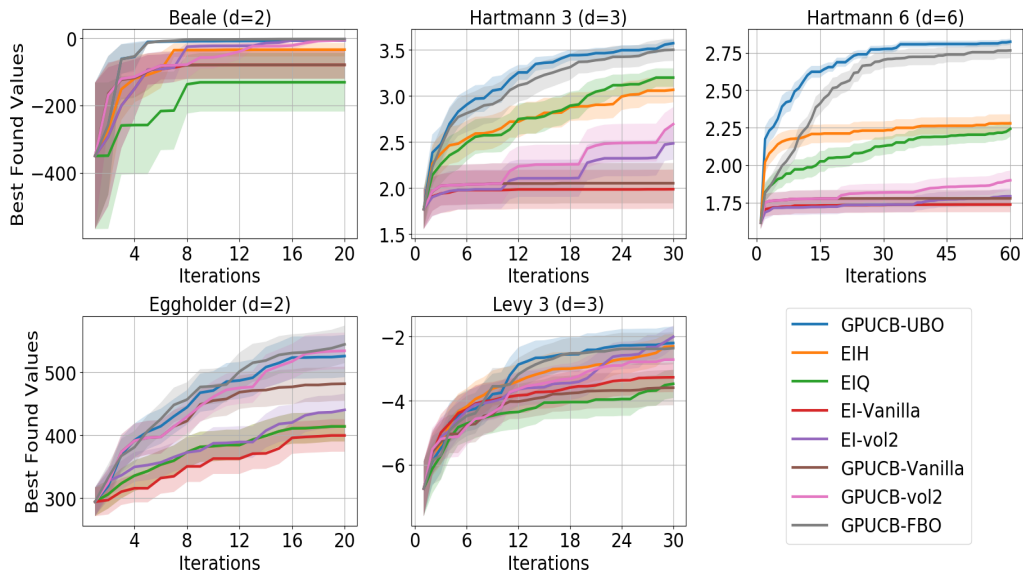

Figure 3: Best found values of various synthetic benchmark test functions using different algorithms. Plotting mean and standard error over 30 repetitions. (Best seen in color)

Table 1: The average runtime (seconds) of selecting the next input of different methods. All the time measurements were taken when evaluating the methods on a Ubuntu 18.04.2 server with Intel Xeon CPU E5-2670 2.60GHz 128GB RAM. All the source codes are written in Python 3.6.

| METHODS | Beale | Eggholder | Hartman3 | Levy3 | Hartman6 |
|---|---|---|---|---|---|
| GPUCB-UBO | $2.8 \pm 0.2$ | $2.8 \pm 0.3$ | $3.1 \pm 0.5$ | $3.7 \pm 0.5$ | $5.0 \pm 0.9$ |
| EIH | $3.4 \pm 0.2$ | $1.0 \pm 0.01$ | $1.2 \pm 0.03$ | $4.9 \pm 0.2$ | $1.4 \pm 0.02$ |
| EIQ | $5.6 \pm 0.4$ | $2.9 \pm 0.02$ | $3.3 \pm 0.03$ | $5.8 \pm 0.3$ | $5.7 \pm 0.1$ |
| EI-Vol2 | $3.2 \pm 0.2$ | $0.9 \pm 0.01$ | $1.2 \pm 0.1$ | $5.1 \pm 0.2$ | $1.7 \pm 0.1$ |
| GPUCB-Vol2 | $3.5 \pm 0.4$ | $1.6 \pm 0.05$ | $9.4 \pm 0.7$ | $2.9 \pm 0.1$ | $12.0 \pm 1.1$ |
| GPUCB-FBO | $5.6 \pm 0.4$ | $5.4 \pm 0.2$ | $8.3 \pm 1.1$ | $8.6 \pm 0.3$ | $18.8 \pm 2.9$ |

## 6.3 Hyperparameter Tuning for Machine Learning Models

Next we apply our method on hyperparameter tuning of three machine learning models on the MNIST dataset: elastic net, multilayer perceptron and convolutional neural network. With each model, the experiments are repeated 20 times and each time the initial search space will be placed differently.

**Elastic Net** Elastic net is a regularized regression method that utilizes the $L_1$ and $L_2$ regularizers. In the model, the hyperparameter $\alpha > 0$ determines the magnitude of the penalty and the hyperparameter $l$ ($0 \le l \le 1$) balances between the $L_1$ and $L_2$ regularizers. We tune $\alpha$ in the normal space while $l$ is tuned in an exponent space (base 10). The initial search space of $\alpha$ and $l$ is randomly placed in the domain $[-3, -1] \times [0, 1]$ with side length to be $20\%$ of the domain size length. We implement the Elastic net model using the function *SGDClassifier* in the scikit-learn package [13].

**Multilayer Perceptron (MLP)** We construct a 2-layer MLP with 512 neurons/layer. We optimize three hypeparameters: the learning rate $l$ and the $L_2$ norm regularization hyperparameters $l_{r1}$ and $l_{r2}$ of the two layers. All the hyperparameters are tuned in the exponent space (base 10). The initial search space is a randomly placed unit cube in the cube $[-6, -1]^3$. The model is implemented using tensorflow. The model is trained with the Adam optimizer in 20 epochs and the batch size is 128.

**Convolutional Neural Network (CNN)** We consider a CNN with two convolutional layers. The CNN architecture (e.g. the number of filters, the filter shape, etc.) is chosen as the standard architecture published on the official GitHub repository of tensorflow [1]. We optimize three hyperparameters: the learning rate $l$ and the dropout rates $r_{d1}, r_{d2}$ in the pooling layers 1 and 2. We tune $r_{d1}, r_{d2}$ in the normal space while $l$ is tuned in an exponent space (base 10). The initial search space of $r_{d1}, r_{d2}, l$ is randomly placed in the domain $[0, 1] \times [0, 1] \times [-5, -1]$ with side length to be $20\%$ of this domain size length. The network is trained with the Adam optimizer in 20 epochs and the batch size is 128.

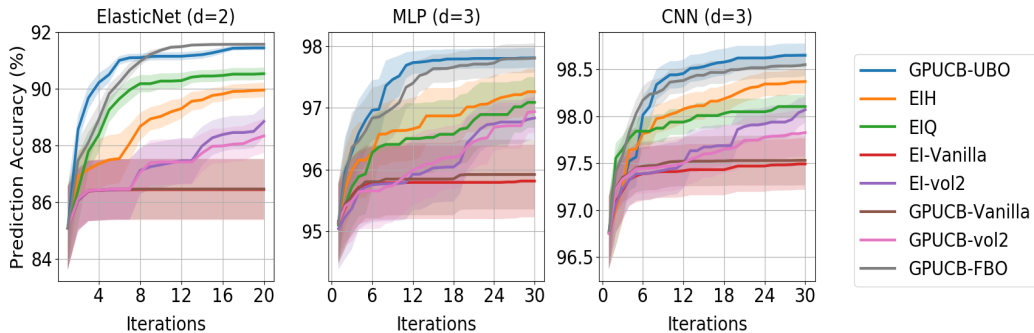

Figure 4: Prediction accuracy of different machine learning models on MNIST dataset using different algorithms. Mean and standard error over 20 repetitions are shown. (Best seen in color)

Given a set of hyperparameters, we train the models with this hyperparameter setting using the MNIST train dataset (55000 patterns) and then test the model on the MNIST test dataset (10000 patterns). Bayesian optimization method then suggests a new hyperparameter setting based on the

[1]https://github.com/tensorflow/tensorflow

prediction accuracy on the test dataset. This process is conducted iteratively until the evaluation budget ($10d$ evaluations) is depleted. We plot the prediction accuracy in Figure 4. For the Elastic net model, our method GPUCB-UBO performs similar to GPUCB-FBO while outperforming the other six approaches significantly. For the MLP model, GPUCB-UBO performs far better than other approaches. To be specific, after only $12$ iterations, it achieves a prediction accuracy of $97.8\%$ whilst other approaches take more than $24$ iterations to get to this level. For the CNN model, GPUCB-UBO also outperforms other approaches by a high margin. After $30$ iterations, it can provide a CNN model with prediction accuracy of $98.7\%$.

# 7 Conclusion

We propose a novel Bayesian optimization framework when the search space is unknown. We guarantee that in iterative expansions of the search space, our method can find a point whose function value within $\epsilon$ of the objective function maximum. Without the need to specify any parameters, our algorithm automatically triggers a minimal expansion required iteratively. We demonstrate our method on both synthetic benchmark functions and machine learning hyper-parameter tuning tasks and demonstrate that our method outperforms state-of-the-art approaches.

Our source code is publicly available at *https://github.com/HuongHa12/BO_unknown_searchspace*.

**Acknowledgments**

This research was partially funded by the Australian Government through the Australian Research Council (ARC). Prof Venkatesh is the recipient of an ARC Australian Laureate Fellowship (FL170100006).

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
