[Supplementary Material · BO_unknownsearchspace_SupplementaryMaterial.pdf]

# Bayesian Optimization with Unknown Search Space - Supplementary Material

**Huong Ha, Santu Rana, Sunil Gupta, Thanh Nguyen, Hung Tran-The, Svetha Venkatesh**
Applied Artificial Intelligence Institute (A$^2$I$^2$)
Deakin University, Geelong, Australia
{huong.ha, santu.rana, sunil.gupta, thanhnt, hung.tranthe, svetha.venkatesh}@deakin.edu.au

In this supplementary material, we include the detailed proofs for the Propositions 4.1, 5.1 and the Theorems 4.1, 5.1. We also provide further details of the search space optimization. Finally, we show extra experimental results on the test functions Levy10 and Ackley10.

## 1 Proof of All the Propositions and Theorems

### 1.1 Background

**Gaussian Process**  A Gaussian Process (GP) is a distribution over functions, which is completely specified by its mean function and covariance function [4]. Assume the function $f$ follows a GP with mean function $m_0(x)$ and covariance function $k(x, x')$. Given the observed data $\mathcal{D}_{t-1} = \{(x_i, y_i)\}_{i=1}^n$, $y_i = f(x_i) + \xi_i$, where $\xi_i \sim \mathcal{N}(0, \sigma^2)$ is the noise, the posterior distribution of $f$ is a GP with the following posterior mean and variance,

$$
\begin{aligned}
\mu_{t-1}(x) &= m_0(x) + \mathbf{k}_{|\mathcal{D}_{t-1}|}(x)^T (\mathbf{K}_{|\mathcal{D}_{t-1}|} + \sigma^2 \mathbf{I}_{|\mathcal{D}_{t-1}|})^{-1} \mathbf{y}_{|\mathcal{D}_{t-1}|}, \\
\sigma_{t-1}^2(x) &= k(x, x) - \mathbf{k}_{|\mathcal{D}_{t-1}|}(x)^T (\mathbf{K}_{|\mathcal{D}_{t-1}|} + \sigma^2 \mathbf{I}_{|\mathcal{D}_{t-1}|})^{-1} \mathbf{k}_{|\mathcal{D}_{t-1}|}(x),
\end{aligned}
\tag{1}
$$

where $\mathbf{y}_{|\mathcal{D}_{t-1}|} = [y_1, \ldots, y_{|\mathcal{D}_{t-1}|}]^T$, $\mathbf{k}_{|\mathcal{D}_{t-1}|}(x) = [k(x, x_i)]_{i=1}^{|\mathcal{D}_{t-1}|}$, $\mathbf{K}_{|\mathcal{D}_{t-1}|} = [k(x_i, x_j)]_{i,j}$, $\mathbf{I}_{|\mathcal{D}_{t-1}|}$ is the $|\mathcal{D}_{t-1}| \times |\mathcal{D}_{t-1}|$ identity matrix and $|\mathcal{D}_{t-1}|$ denotes the cardinality of $\mathcal{D}_{t-1}$. To aid readability, in the sequel we remove the notation that shows the dependence of $\mathbf{k}, \mathbf{K}, \mathbf{I}, \mathbf{y}$ on $|\mathcal{D}_{t-1}|$.

**GP-UCB Acquisition Function**  The GP-UCB acquisition function is defined as [1–3, 5],

$$
\alpha_{UCB}(x; \mathcal{D}_{t-1}) = \mu_{t-1}(x) + \sqrt{\beta_t} \sigma_{t-1}(x),
\tag{2}
$$

where $\mu_{t-1}(x), \sigma_{t-1}(x)$ are the posterior mean and standard deviation of the GP given observed data $\mathcal{D}_{t-1}$ and $\beta_t \geq 0$ is an appropriate parameter that balances the exploration and exploitation. Given a search domain, $\{\beta_t\}$ can be chosen following the suggestion in [5] to ensure global convergence in this domain.

**Maximum Information Gain**  For any search space $\mathcal{S}$, define the maximum mutual information $\gamma_{T,\mathcal{S}}$ [5]:

$$
\gamma_{T,\mathcal{S}} := \max_{A \subset \mathcal{S}, |A|=T} \frac{1}{2} \log \det(I_T + \sigma^{-2} K_T),
\tag{3}
$$

where $K_T = [k(x, x')]_{x,x' \in A}$ and $I_T$ is the identity matrix with size $T \times T$.

**Defining $g_k(\gamma)$ for kernel k(.)**  With the types of kernels satisfied Assumption 4.2 in the paper, for all small positive $\gamma$, there always exists $g_k(\gamma) > 0$ such that,

$$
\forall x, x' : \|x - x'\|_2 \geq g_k(\gamma), \quad k(x, x') \leq \gamma.
\tag{4}
$$

The value of $g_k(\gamma)$ can be computed from $\gamma$ and the kernel covariance function $k(x, x')$ i.e. for Squared Exponential kernel $k_{SE}(x, x') = \theta^2 \exp(-\|x - x'\|_2^2/(2l^2))$, $g_k(\gamma)$ will be $\sqrt{2l^2 \log(\theta^2/\gamma)}$.

## 1.2 Proof of Proposition 4.1

Substituting Eq. (1) into Eq. (2) and combining with Assumption 4.1 in the paper, we have,

$$\alpha_{UCB}(x; \mathcal{D}_{t-1}) = \mathbf{k}(x)^T (\mathbf{K} + \sigma^2 \mathbf{I})^{-1} \mathbf{y} + \sqrt{\beta_t} \sqrt{k(x,x) - \mathbf{k}(x)^T (\mathbf{K} + \sigma^2 \mathbf{I})^{-1} \mathbf{k}(x)}. \tag{5}$$

Since $k(x, x')$ is a kernel covariance function that satisfies the Assumptions 4.2 in the paper, then,

$$\mathbf{k}(x) \xrightarrow{x \to \infty} \mathbf{0}, \quad k(x,x) = \theta^2. \tag{6}$$

Combining Eq. (5) and Eq. (6), the proposition is proved, i.e.,

$$\alpha_{UCB}(x; \mathcal{D}_{t-1}) \xrightarrow{x \to \infty} \sqrt{\beta_t}\theta. \qquad \blacksquare$$

## 1.3 Proof of Theorem 4.1

Firstly, we prove that with the choice of $\mathcal{S} = \mathbb{R}^d \setminus \mathcal{A}$, where 1) $\mathcal{A}$ contains all the points $x$ that are far from all the current observations, and, 2) $\mathcal{A} := \{x \in \mathbb{R}^d : |\alpha_{UCB}(x; \mathcal{D}_{t-1}) - \sqrt{\beta_t}\theta| < \epsilon/2\}$, there exists a point in $\mathcal{S}$ whose GP-UCB acquisition function value is within $\epsilon$ from the maximum of the acquisition function, i.e. $\exists x_u \in \mathcal{S} : |\alpha_{UCB}(x_u; \mathcal{D}_{t-1}) - \max_{x \in \mathbb{R}^d} \alpha_{UCB}(x; \mathcal{D}_{t-1})| \leq \epsilon$. With the choice of $\epsilon < |\sqrt{\beta_t}\theta - \min_{x \in \mathbb{R}^d}(\alpha_{UCB}(x; \mathcal{D}_{t-1}))|$, let us consider three cases:

- **Case 1**: The argmax of the GP-UCB acquisition function is at infinity. This means that the GP-UCB acquisition function maximum is equal to $\sqrt{\beta_t}\theta$. As the GP-UCB acquisition function is continuous and $\epsilon < |\sqrt{\beta_t}\theta - \min_{x \in \mathbb{R}^d}(\alpha_{UCB}(x; \mathcal{D}_{t-1}))|$, hence, there exists a point $x_u$ such that $\alpha_{UCB}(x_u) = \sqrt{\beta_t}\theta - \epsilon/2$. By the definition of $\mathcal{S}$, it is straightforward that $x_u$ belongs to $\mathcal{S}$, thus proving that there exists a point in $\mathcal{S}$ whose GP-UCB acquisition function value is within $\epsilon$ from the maximum of the acquisition function.

- **Case 2**: The argmax of the GP-UCB acquisition function $x'_{max}$ is at a finite location and its acquisition function value is larger or equal $\sqrt{\beta_t}\theta + \epsilon/2$. It is straightforward to see that the argmax $x'_{max}$ belongs to the region $\mathcal{S}$ and this is the point that satisfies $|\alpha_{UCB}(x'_{max}; \mathcal{D}_{t-1}) - \max_{x \in \mathbb{R}^d} \alpha_{UCB}(x; \mathcal{D}_{t-1})| \leq \epsilon$.

- **Case 3**: The GP-UCB acquisition function argmax is at a finite location and the acquisition function maximum is smaller than $\sqrt{\beta_t}\theta + \epsilon/2$. As the GP-UCB acquisition function is continuous and $\epsilon < |\sqrt{\beta_t}\theta - \min_{x \in \mathbb{R}^d}(\alpha_{UCB}(x; \mathcal{D}_{t-1}))|$, there exists a point $x_u \in \mathcal{S}$ : $\alpha_{UCB}(x_u; \mathcal{D}_{t-1}) = \sqrt{\beta_t}\theta - \epsilon/2$. As $\max_{x \in \mathbb{R}^d} \alpha_{UCB}(x; \mathcal{D}_{t-1}) < \sqrt{\beta_t}\theta + \epsilon/2$, it follows directly that $|\alpha_{UCB}(x_u; \mathcal{D}_{t-1}) - \max_{x \in \mathbb{R}^d} \alpha_{UCB}(x; \mathcal{D}_{t-1})| \leq \epsilon$.

Secondly, we will give analytical expression for one way to define the region $\mathcal{A}$. We will prove that $\mathcal{A}$ can be chosen as $\{x : \forall x_i \in \mathcal{D}_{t-1} : \|x - x_i\|_2 \geq g_k(\gamma)\}$, where,

$$\gamma = \min(\sqrt{(0.5\sqrt{\beta_t}\theta\epsilon - 0.0625\epsilon^2)/(|\mathcal{D}_{t-1}|\lambda_{max})}\theta, 0.25\epsilon/\max\Big(\sum_{z_j \leq 0} -z_j, \sum_{z_j \geq 0} z_j\Big)),$$

$g_k(.)$ defined as in Eq. (4), $\lambda_{max} \in \mathbb{R}^+$ is the largest singular value of $(\mathbf{K} + \sigma^2 \mathbf{I})^{-1}$, $z_j$ is the $j^{th}$ element of vector $(\mathbf{K} + \sigma^2 \mathbf{I})^{-1} \mathbf{y}$ and $\epsilon < |\sqrt{\beta_t}\theta - \min_{x \in \mathbb{R}^d}(\alpha_{UCB}(x; \mathcal{D}_{t-1}))|$. With this choice of $\mathcal{A}$, then the region $\mathcal{S}$ can be computed as $\mathcal{S} = \bigcup_{i=1}^{|\mathcal{D}_{t-1}|} \mathcal{S}_i$, $\mathcal{S}_i = \{x : \|x - x_i\|_2 \leq g_k(\gamma)\}, x_i \in \mathcal{D}_{t-1}$.

Consider the GP-UCB acquisition function,

$$\begin{aligned}
\alpha_{UCB}(x; \mathcal{D}_{t-1}) &= \mu_{t-1}(x) + \sqrt{\beta_t}\sigma_{t-1}(x) \\
&= \mathbf{k}(x)^T (\mathbf{K} + \sigma^2 \mathbf{I})^{-1} \mathbf{y} + \sqrt{\beta_t}\sqrt{k(x,x) - \mathbf{k}(x)^T (\mathbf{K} + \sigma^2 \mathbf{I})^{-1} \mathbf{k}(x)} \\
&= \mathbf{k}(x)^T (\mathbf{K} + \sigma^2 \mathbf{I})^{-1} \mathbf{y} + \sqrt{\beta_t}\sqrt{\theta^2 - \mathbf{k}(x)^T (\mathbf{K} + \sigma^2 \mathbf{I})^{-1} \mathbf{k}(x)}.
\end{aligned}$$

The error between $\alpha_{UCB}(x; \mathcal{D}_{t-1})$ and $\sqrt{\beta_t}\theta$ can be computed as,

$$\sqrt{\beta_t}\theta - \alpha_{UCB}(x; \mathcal{D}_{t-1}) = -\mathbf{k}(x)^T (\mathbf{K} + \sigma^2 \mathbf{I})^{-1} \mathbf{y} + \frac{\sqrt{\beta_t}\mathbf{k}(x)^T (\mathbf{K} + \sigma^2 \mathbf{I})^{-1} \mathbf{k}(x)}{\sqrt{\theta^2 - \mathbf{k}(x)^T (\mathbf{K} + \sigma^2 \mathbf{I})^{-1} \mathbf{k}(x)} + \theta}. \tag{7}$$

First, we consider the first term in Eq. (7). It is easy to see that,

$$\sum_{z_i \leq 0} z_i k(x, x_i) \leq \mathbf{k}(x)^T (\mathbf{K} + \sigma^2 \mathbf{I})^{-1} \mathbf{y} \leq \sum_{z_i > 0} z_i k(x, x_i),$$

where $z_i$ is the $i^{th}$ element of vector $(\mathbf{K} + \sigma^2 \mathbf{I})^{-1} \mathbf{y}$. Therefore,

$$|\mathbf{k}(x)^T (\mathbf{K} + \sigma^2 \mathbf{I})^{-1} \mathbf{y}| \leq \max \left( \left| \sum_{z_i \leq 0} z_i k(x, x_i) \right|, \left| \sum_{z_i \geq 0} z_i k(x, x_i) \right| \right)$$

$$\leq \max \left( \sum_{z_i \leq 0} -z_i k(x, x_i), \sum_{z_i \geq 0} z_i k(x, x_i) \right). \tag{8}$$

For all small positive $\gamma$, for all $x$ such that $\|x - x_i\|_2 \geq g_k(\gamma), \forall x_i \in \mathcal{D}_{t-1}$ with $g_k(\gamma)$ defined in Eq. (4), we have $k(x, x_i) \leq \gamma, \forall x_i \in \mathcal{D}_{t-1}$. Combining this with Eq. (8), the first term in Eq. (7) is bounded as,

$$|\mathbf{k}(x)^T (\mathbf{K} + \sigma^2 \mathbf{I})^{-1} \mathbf{y}| \leq \max \left( \sum_{z_i \leq 0} -z_i, \sum_{z_i \geq 0} z_i \right) \gamma. \tag{9}$$

Second, consider the second term in Eq. (7). Since $\mathbf{K}$ is a covariance matrix, it is a positive semidefinite matrix, hence, $(K + \sigma^2 I)$ and $(K + \sigma^2 I)^{-1}$ are also positive semidefinite symmetric matrices. Using the singular value decomposition (SVD), we have, $(K + \sigma^2 I)^{-1} = UDU^T$, where $D$ is a diagonal matrix and $U$ is a unitary matrix. Denote $\lambda_{max}$ ($\lambda_{max} \in \mathbb{R}^+$) to be the maximum entry on the diagonal of matrix $D$ ($\lambda_{max}$ is also called the largest singular value of the matrix $(K + \sigma^2 I)^{-1}$), we then have,

$$\mathbf{k}(x)^T ((\mathbf{K} + \sigma^2 \mathbf{I})^{-1} - \lambda_{max} \mathbf{I}) \mathbf{k}(x) = \mathbf{k}(x)^T (UDU^T - U\lambda_{max} \mathbf{I} U^T) \mathbf{k}(x)$$

$$= (U^T \mathbf{k}(x))^T (D - \lambda_{max} \mathbf{I}) U^T \mathbf{k}(x).$$

Since $(D - \lambda_{max} \mathbf{I})$ is a negative semidefinite matrix, $(U^T \mathbf{k}(x))^T (D - \lambda_{max} \mathbf{I}) U^T \mathbf{k}(x) \leq 0$. Therefore,

$$\mathbf{k}(x)^T (\mathbf{K} + \sigma^2 \mathbf{I})^{-1} \mathbf{k}(x) \leq \lambda_{max} \mathbf{k}(x)^T \mathbf{k}(x) \leq \lambda_{max} \sum_{i=1}^{|\mathcal{D}_{t-1}|} k^2(x, x_i). \tag{10}$$

For all small positive $\gamma$, for all $x$ such that $\|x - x_i\|_2 \geq g_k(\gamma), \forall x_i \in \mathcal{D}_{t-1}$ with $g_k(\gamma)$ defined in Eq. (4), we have $k(x, x_i) \leq \gamma, \forall x_i \in \mathcal{D}_{t-1}$. Combining this with Eq. (10), we now have,

$$0 \leq \mathbf{k}(x)^T (\mathbf{K} + \sigma^2 \mathbf{I})^{-1} \mathbf{k}(x) \leq n\lambda_{max} \gamma^2, \tag{11}$$

where $n$ denotes $|\mathcal{D}_{t-1}|$, i.e. cardinality of $\mathcal{D}_{t-1}$. Consider the function,

$$f(z) = \frac{\sqrt{\beta_t} z}{\sqrt{\theta^2 - z} + \theta}, \quad 0 \leq z \leq \theta^2.$$

It is easy to see $f(z)$ is a monotone increasing function in the range $[0, \theta^2]$. Hence, with $\mathbf{k}(x)^T (\mathbf{K} + \sigma^2 \mathbf{I})^{-1} \mathbf{k}(x)$ being in the range $[0, n\lambda_{max} \gamma^2]$, then,

$$0 \leq \frac{\sqrt{\beta_t} \mathbf{k}(x)^T (\mathbf{K} + \sigma^2 \mathbf{I})^{-1} \mathbf{k}(x)}{\sqrt{\theta^2 - \mathbf{k}(x)^T (\mathbf{K} + \sigma^2 \mathbf{I})^{-1} \mathbf{k}(x)} + \theta} \leq \frac{\sqrt{\beta_t} n\lambda_{max} \gamma^2}{\sqrt{\theta^2 - n\lambda_{max} \gamma^2} + \theta}, \tag{12}$$

where $\gamma \leq \sqrt{\theta/(n\lambda_{max})}$. From Eqs. (7), (9) and (12), we have that: $\forall \gamma > 0$, then $\forall x$ such that $\|x - x_i\|_2 \geq d_\gamma, i = \overline{1, n}$, the following inequality is satisfied,

$$|\sqrt{\beta_t} \theta - \alpha_{UCB}(x; \mathcal{D}_{t-1})| \leq \max \left( \sum_{z_i \leq 0} -z_i, \sum_{z_i \geq 0} z_i \right) \gamma + \frac{\sqrt{\beta_t} n\lambda_{max} \gamma^2}{\sqrt{\theta^2 - n\lambda_{max} \gamma^2} + \theta}. \tag{13}$$

With the choice of $\gamma = \min \left( \frac{0.25\epsilon}{\max(\sum_{z_i \leq 0} -z_i, \sum_{z_i \geq 0} z_i)}, \frac{1}{\sqrt{\beta_t}} \sqrt{\frac{0.5\sqrt{\beta_t} \theta\epsilon - 0.0625\epsilon^2}{n\lambda_{max}}} \right), 0 < \epsilon < 4\sqrt{\beta_t} \theta$. Then from Eq. (13), we have that $\forall x : \|x - x_i\|_2 \geq d_\gamma, i = \overline{1, n}$, the following inequality

satisfies,

$$|\sqrt{\beta_t}\theta - \alpha_{UCB}(x;\mathcal{D}_{t-1})| < \max\Big(\sum_{z_i\leq 0}-z_i, \sum_{z_i\geq 0}z_i\Big)\frac{0.25\epsilon}{\max(\sum_{z_i\leq 0}-z_i,\sum_{z_i\geq 0}z_i)}$$

$$+ \frac{\sqrt{\beta_t}n\lambda_{max}\left(\frac{1}{\sqrt{\beta_t}}\sqrt{\frac{0.5\sqrt{\beta_t}\theta\epsilon - 0.0625\epsilon^2}{n\lambda_{max}}}\right)^2}{\sqrt{\theta^2 - n\lambda_{max}\left(\frac{1}{\sqrt{\beta_t}}\sqrt{\frac{0.5\sqrt{\beta_t}\theta\epsilon - 0.0625\epsilon^2}{n\lambda_{max}}}\right)^2} + \theta} \qquad (14)$$

$$< 0.25\epsilon + 0.25\epsilon$$
$$< \epsilon/2. \qquad \blacksquare$$

**Remark 1.1** *Note that if $|\sqrt{\beta_t}\theta - \min_{x\in\mathbb{R}^d}\alpha_{UCB}(x;\mathcal{D}_{t-1})| = 0$, then $\forall \epsilon < |\sqrt{\beta_t}\theta - \max_{x\in\mathbb{R}^d}\alpha_{UCB}(x;\mathcal{D}_{t-1})|$ or $< |\sqrt{\beta_t}\theta - \max_{x\in\mathcal{D}_{t-1}}\alpha_{UCB}(x;\mathcal{D}_{t-1})|$, the bound in Theorem 4.1 remains valid. As in this case, the GP-UCB argmax is at a finite location and its acquisition function value $> \sqrt{\beta_t}\theta$, thus our arguments in Case 2 hold. However, it is worth noting that the scenario $|\sqrt{\beta_t}\theta - \min_{x\in\mathbb{R}^d}\alpha_{UCB}(x;\mathcal{D}_{t-1})| = 0$ is very rare in practice. With Assumption 4.1, most of the time, $\min_{x\in\mathbb{R}^d}\alpha_{UCB}(x;\mathcal{D}_{t-1}) \leq 0$, hence $|\sqrt{\beta_t}\theta - \min_{x\in\mathbb{R}^d}\alpha_{UCB}(x;\mathcal{D}_{t-1})| > 0$. Only when the noise is large, there is a very small chance $|\sqrt{\beta_t}\theta - \min_{x\in\mathbb{R}^d}\alpha_{UCB}(x;\mathcal{D}_{t-1})| = 0$ can happen.*

### 1.4 Proof of Proposition 5.1

Following Lemma 5.7 in [5], for any $d$-dimensional domain $\mathcal{S}_k$ with side length $r_k$, suppose the kernel $k(x,x')$ satisfies the following condition on the derivatives of GP sample paths $f$: $\exists a_k, b_k > 0$, $\Pr\{\sup_{x\in\mathcal{S}_k}|\partial f/\partial x_j| > L\} \leq a_k\exp^{-(L/b_k)^2}, j = \overline{1,d}$. Pick $\delta \in (0,1)$, and define $\beta_t = 2\log(t^2 2\pi^2/(3\delta)) + 2d\log(t^2 db_k r_k\sqrt{\log(4da_k/\delta)})$, then $\forall\epsilon > 0$, with probability larger than $1-\delta$, we have $\max_{x\in\mathcal{S}_k}\alpha_{UCB}(x;\mathcal{D}_{t-1}) \leq \mu_{t-1}(x_t) + \beta_t^{1/2}\sigma_{t-1}(x_t) + 1/t^2, \forall t \geq 1$, where $x_t$ is the suggestion from the GP-UCB algorithm at iteration $t$. Therefore,

$$\max_{x\in\mathcal{S}_k}f(x) - \max_{x\in\mathcal{D}_t}f(x)$$
$$\leq \mu_{t-1}(x_t) + \beta_t^{1/2}\sigma_{t-1}(x_t) + 1/t^2 - \max_{x\in\mathcal{D}_t}f(x)$$
$$\leq \mu_{t-1}(x_t) + \beta_t^{1/2}\sigma_{t-1}(x_t) + 1/t^2 - \max_{x\in\mathcal{D}_t}\alpha_{LCB}(x;\mathcal{D}_{t-1}) \qquad (15)$$
$$\leq \mu_{t-1}(x_t) + \beta_t^{1/2}\sigma_{t-1}(x_t) + 1/t^2 - \alpha_{LCB}(x_t;\mathcal{D}_{t-1})$$
$$\leq 2\beta_t^{1/2}\sigma_{t-1}(x_t) + 1/t^2.$$

Following Lemma 5.4 in [5], we have that,

$$\sum_{t=1}^{T}4\beta_t\sigma_{t-1}^2(x_t) \leq C_1\beta_T\gamma_T, \qquad (16)$$

with $C_1 = 8/\log(1+\sigma^{-2})$, $\gamma_T$ is the maximum information gain in the search space $\mathcal{S}_k$ (can be computed using (3)).

Assume $2\beta_t^{1/2}\sigma_{t-1}(x_t)$ does not converge to 0 when $t \to \infty$. It means there exists $T_0$, such that $\forall t \geq T_0, 2\beta_t^{1/2}\sigma_{t-1}(x_t) \geq m$ with $m$ being a constant. Then, $\forall T \geq T_0$,

$$\sum_{t=T_0}^{T}4\beta_t\sigma_{t-1}^2(x_t) \geq m^2(T-T_0). \qquad (17)$$

Which means, $\forall T \geq T_0$,

$$C_1\beta_T\gamma_T - \sum_{t=1}^{T_0}4\beta_t\sigma_{t-1}^2(x_t) \geq m^2(T-T_0). \qquad (18)$$

For the kernel classes: finite dimensional linear, Squared Exponential and Matérn, and assume the kernel satisfies $k(x, x') \leq 1$ (condition 2 of Assumption 4.2 in the paper), we have that $\gamma_T$ is upper bounded by $\mathcal{O}(T^{d(d+1)/(2\nu+d(d+1))}(\log T))$ with $\nu > 1$. However, the RHS of Eq. (18) is $\mathcal{O}(T)$, thus it is not correct. Therefore, $2\beta_t^{1/2}\sigma_{t-1}(x_t)$ converges to 0 when $t \to \infty$. Which means there $\exists T_k : \forall t \geq T_k$,

$$r_{b,\mathcal{S}_k}(t) = \max_{x \in \mathcal{S}_k} \alpha_{UCB}(x; \mathcal{D}_{t-1}) - \max_{x \in \mathcal{D}_t} \alpha_{LCB}(x; \mathcal{D}_{t-1}) \leq \epsilon - 1/t^2. \qquad (19)$$

Finally, following Eq. (15), with probability larger than $1 - \delta$, we have, $\forall t$, $\max_{x \in \mathcal{S}_k} \alpha_{UCB}(x; \mathcal{D}_{t-1})$
$- \max_{x \in \mathcal{D}_t} f(x) \leq \mu_{t-1}(x_t) + \beta_t^{1/2}\sigma_{t-1}(x_t) + 1/t^2 - \max_{x \in \mathcal{D}_t} \alpha_{LCB}(x; \mathcal{D}_{t-1})$. Thus $\forall t$ satisfies Eq. (19), $\max_{x \in \mathcal{S}_k} f(x) - \max_{x \in \mathcal{D}_t} f(x) \leq \epsilon$. ∎

## 1.5 Proof of Theorem 5.1

First, we prove that with our search space expansion strategy, the search space will continue to expand, i.e. size$(\mathcal{S}_k) \xrightarrow{k\to\infty} \infty$. As $\lambda_{max}$ is the largest singular value of the matrix $(K + \sigma^2 I)^{-1}$, hence, it is also the maximum entry on the diagonal of matrix $D$ where $D$ satisfies: $(K + \sigma^2 I)^{-1} = U D U^T$ with $U$ being a unitary matrix. Note that $\lambda_{max} = 1/(\lambda_{min,(K+\sigma^2 I)})$ where $\lambda_{min,(K+\sigma^2 I)}$ is the smallest singular value of matrix $(K + \sigma^2 I)$. We have,

$$n\lambda_{min,(K+\sigma^2 I)} \leq \text{Tr}(K + \sigma^2 I) = n(\theta^2 + \sigma^2), \qquad (20)$$

where Tr(.) is the Trace operator and $n = |\mathcal{D}_{t-1}|$. Thus $\lambda_{min,(K+\sigma^2 I)} \leq (\theta^2 + \sigma^2)$, which results $\lambda_{max} \geq 1/(\theta^2 + \sigma^2)$. Therefore,

$$\sqrt{(0.5\sqrt{\beta_t}\theta\epsilon - 0.0625\epsilon^2)/(|\mathcal{D}_{t-1}|\lambda_{max})}\theta \xrightarrow{t\to\infty} 0, \qquad (21)$$

as $\beta_t \sim \mathcal{O}(\log(t))$ and $|\mathcal{D}_{t-1}| \geq t - 1$.

This means $d_\epsilon \xrightarrow{t\to\infty} \infty$, hence size$(\mathcal{S}_k) \xrightarrow{k\to\infty} \infty$. Thus $\exists k_0$ such that $\mathcal{S}_* \subset \mathcal{S}_{k_0}$, where $\mathcal{S}_*$ is the region that contains the objective function global maximum. Following Proposition 5.1 in the paper, with the choice of $\{\beta_t\}$ to be as suggested in the Theorem, the proposed algorithm achieves the *local $\epsilon$-accuracy* any search space. Thus it eventually achieves the *local $\epsilon$-accuracy* in search space $\mathcal{S}_{k_0}$, hence $\exists x_{k_0} : (f(x_{k_0}) - \max_{x \in \mathcal{S}_0} f(x)) \leq \epsilon$. Note that $\max_{x \in \mathcal{S}_0} f(x) = \max_{x \in \mathcal{S}_*} f(x)$ as $\mathcal{S}_* \subset \mathcal{S}_{k_0}$. Therefore, $(\max_{x \in \mathcal{S}_*} f(x) - f(x_{k_0})) \leq \epsilon$. ∎

# 2 Further Details of Search Space Optimization

The theoretical search space developed in Theorem 4.1 is the union of $|\mathcal{D}_{t-1}|$ balls. To suit optimizer input, this region is converted to an encompassing hypercube using,

$$\min_{x_i \in \mathcal{D}_{t-1}}(x_i^k) - d_\epsilon \leq x^k \leq \max_{x_i \in \mathcal{D}_{t-1}}(x_i^k) + d_\epsilon, \ k = \overline{1, d}. \qquad (22)$$

This encompassing hypercube is larger than our theoretical search space. In the case when the GP-UCB acquisition function argmax is at finite location, then acquisition function maximum within our theoretical search space and within this encompassing hypercube is the same. In the case when the GP-UCB acquisition function argmax is at infinity, optimizing the acquisition function within the encompassing hypercube results in the suggested point to be much further than it should be. Therefore, in our algorithm, after optimizing the GP-UCB acquisition function within the encompassing hypercube, we check,

- If the GP-UCB acquisition function maximum is larger than $\sqrt{\beta_t}\theta$ or smaller than $\sqrt{\beta_t}\theta - \epsilon$, then suggest that point as the interesting point to be evaluated.

- If the GP-UCB acquisition function maximum is between $\sqrt{\beta_t}\theta - \epsilon$ and $\sqrt{\beta_t}\theta$, then construct $|\mathcal{D}_{t-1}|$ encompassing hypercubes for the $|\mathcal{D}_{t-1}|$ balls. To save computation time, optimize the acquisition function within the hypercube that encompass the ball whose center has the highest $\alpha_{UCB}$ value first. Then check if the acquisition function maximum is smaller than $\sqrt{\beta_t}\theta - \epsilon$, if yes, stop and pick that point to be the interesting point to be evaluated. Otherwise, continue until find one.

# 3 Extra Experimental Results

In this section, we show extra experimental results on the test functions Levy10 and Ackley10 that we provide NeurIPS reviewers during rebuttal. The setups are same as in the paper. For Ackley10, the number of experiments is 30 whilst for Levy10, the number of experiments is 10 as GPUCB-FBO computation time for Levy10 is so expensive that we can only get 10 experiments during the whole rebuttal time. For Ackley10, our proposed method outperforms other 6 methods by a high margin and is better than GPUCB-FBO and, note that GPUCB-FBO computation time is at least 5-6 times slower than our method. For Levy10, our proposed method is slightly better than EIH, EI-vol2 while outperforming other baselines significantly.

Figure 1: Best found values of two high-dimensional synthetic benchmark test functions using different algorithms. Function Ackley10 is plotted over 30 repetitions whilst Levy10 is plotted over 10 repetitions due to prohibitive computation time of the method GPUCB-FBO. (Best seen in color)