[Reviews · NeurIPS 2019]

Reviewer 1



Applying Bayesian optimization to expensive black-box problems needs to specify the bound of search space. However, when tackling a completely new problem, there is no prior knowledge to guarantee that the specified search space contains the global optimum. The paper proposes an approach to deal with this situation. In the approach, the user first specifies an initial search space; then the bound of search space automatically expands as the iteration proceeds; finally the algorithm will return a solution achieving \epsilon-accuracy. The key is how to expand the search space. The approach uses upper confidence bound (UCB) as acquisition function, and selects the expanded search space containing at least one point whose acquisition function value is within \epsilon from the acquisition function maximum. Also because the gap between the best selected point and the global optimum of the current search space can be bounded, the approach can return a solution achieving \epsilon-accuracy. I have checked all the proofs, and believe most of them are correct. However, I also found some places which are unclear and may be incorrect. Questions: 1. In the proof of Theorem 4.1, \epsilon is set less than \| \sqrt(\beta_t)\theta -\min_{x\in R^d}a_{UCB}(x;D_{t-1})\|. What if \| \sqrt(\beta_t)\theta -\min_{x\in R^d}a_{UCB}(x;D_{t-1})\|=0? 2. \gamma is the minimum value among a set of numbers above eq(14), but is a specific value in eq(14). How do you determine its value in eq(14)? 3. Why do the second and the third inequalities hold in eq(15)? 4. How can eq(15) hold with probability at least 1-\delta? max_{x\in D_t}a_{LCB} is used to replace max_{x\in D_t}f(x) in the second inequality. Did you consider the probability for max_{x\in D_t}a_{LCB}<= max_{x\in D_t}f(x) to hold? Question about experiments: 1. a_k and b_k in \beta_t are unknown. What are their values in your experimental settings? Minor comments: 1. The posterior mean in eq(1) is not correct. “y” -> “y-m”? 2. Proposition 5.1 is proposed in the paper, but it is presented as a lemma in the supplementary.

Reviewer 2



This paper proposes an algorithm to expand the search space for Bayesian Optimization in order to find an optima giving us some guarantees. The paper is complete in the sense that it provides theoretical claims to support their evidence, a proposition of an algorithm and some experiments Although this is nice is another extension of UCB (they are a lot since it is nice to play with it from the theoretical point of view) and this task of expanding the search space has been treated in a lot of NIPS papers this year, with other strategies that I believe that are more practical and general. Related to: The UCB acquisition function, \epsilon accuracy. Strengths: Provides theoretical and empirical content. Weaknesses: Not the most practical approach to perform this task. It is a combination of well known techniques. Some clarity and organization. Does this submission add value to the NIPS community? : Although other approaches perform similar tasks this is another view that might be taken into account. Perhaps it is not the most practical one although. Quality: Is this submission technically sound?: Not a lot, but what is exposes is sound in the sense that I have not read the \epsilon accuracy in BO before. Are claims well supported by theoretical analysis or experimental results?: Yes they are, it is a strength of the paper. Is this a complete piece of work or work in progress?: I would say that it is a complete piece of work. Are the authors careful and honest about evaluating both the strengths and weaknesses of their work?: Yes, I believe. Clarity: Is the submission clearly written?: I have some suggestions that may be corrected in my humble opinion. Is it well organized?: Yes it is. Does it adequately inform the reader?: I believe it to be so. Originality: Are the tasks or methods new?: They already exist in their literature. Is the work a novel combination of well-known techniques?: Combination of well-known techniques. Is it clear how this work differs from previous contributions?: Yes, but not so clear to add real value in practice. Is related work adequately cited?: Yes it is with one exception. Significance: Are the results important?: I would not bet that this approach is going to be used massively in practice. Are others likely to use the ideas or build on them?: I do not know, maybe, but I like other expansion approaches proposed to NIPS this year more. Does the submission address a difficult task in a better way than previous work?: Previous maybe, current no. Does it advance the state of the art in a demonstrable way?: Yes, they provide experiments and theoretical content. Does it provide unique data, unique conclusions about existing data, or a unique theoretical or experimental approach?: Yes, their work is genuine. Arguments for acceptance: It is a complete paper with empirical and theoretical content. Arguments against acceptance: Other space expansions approaches proposed to NIPS this year are more general and practical, maybe there is no room in NIPS for all of them. Being NIPS the best ML conference, the quality of the paper may be borderline. If it were other conference I would recommend this paper for acceptance blindly. Typos: -> I would not put \epsilon accuracy in the abstract, it it too technical. -> Phrase 3 in the abstract is not syntactically correct "our method can find a point whose function value within \epsilon of the objective..." -> Second phrase of introduction is also not syntactically correct. -> I miss a formal definition of the problem in the intro. -> Global \epsilon accuracy must be explained, introduced formally or cited. -> Sections of the paper must be introduced at the end of section 1. -> Problem setup assumes maximization. More detailed comments and suggestions: I would like to congratulate the authors for the paper and recommend it for acceptance with a weak accept (6) because I think that it is a good paper but NIPS has very high standards and it is a pity to say that the other papers that I have reviewed that achieve this same task are more practical, more pragmatic. I would use them in practice and I am afraid that this one not. This is the reason why I qualify this paper with a 6.

Reviewer 3



The current version provides limited explanation and analysis of its empirical results and feels a bit preliminary now. The assumptions and implications of the bounds are not discussed. The theoretical choice of {beta_t} in Theorem 5.1 is typically overly conservative and thus, the practical schedule is used. This is common to GP-UCB based algorithms and should be criticized more because the theoretical analysis and the practical usage is different. In particular, it is more important to select the schedule of beta_t. My main concern is the experimental results. You only analyzed the low-dimensional case (d=2-3) in real problems with the small number of iterations. Optimization should be inefficient if you expand space. Extending the search space seems to need more trails of optimization. When you use the high-dimensional case (d=10) and more iterations case, e.g., 100 iterations, the existing method can outperform your method. Using warped Gaussian process in input space is an approach to reconstruct search space, which is a limited input search space but warped by bijective transformations. Snoek+, INPUT WARPING FOR BAYESIAN OPTIMIZATION OF NON-STATIONARY FUNCTIONS

[Author Response · NeurIPS 2019]

We thank the reviewers for their constructive comments.

————-REVIEWER 1————-

*# What if $|\sqrt{\beta_t}\theta - \min_{x\in\mathbb{R}^d}\alpha_{UCB}(x;\mathcal{D}_{t-1})| = 0$ ?*

If $|\sqrt{\beta_t}\theta - \min_{x\in\mathbb{R}^d}\alpha_{UCB}(x;\mathcal{D}_{t-1})| = 0$, then $\forall\epsilon < |\sqrt{\beta_t}\theta - \max_{x\in\mathbb{R}^d}\alpha_{UCB}(x;\mathcal{D}_{t-1})|$ or $< |\sqrt{\beta_t}\theta -$

$\max_{x\in\mathcal{D}_{t-1}}\alpha_{UCB}(x;\mathcal{D}_{t-1})|$, the bound in Theorem 4.1 remains valid. As in this case, the GP-UCB argmax is at a finite

location and its acquisition function value $> \sqrt{\beta_t}\theta$, thus our arguments in Case 2 of Section 1.3 in the Supplementary ma-

terial hold. However, it is worth noting that the scenario $|\sqrt{\beta_t}\theta - \min_{x\in\mathbb{R}^d}\alpha_{UCB}(x;\mathcal{D}_{t-1})| = 0$ is very rare in practice.

With Assumption 4.1, most of the time, $\min_{x\in\mathbb{R}^d}\alpha_{UCB}(x;\mathcal{D}_{t-1}) \leq 0$, hence $|\sqrt{\beta_t}\theta - \min_{x\in\mathbb{R}^d}\alpha_{UCB}(x;\mathcal{D}_{t-1})| > 0$.

Only when the noise is large, there is a very small chance $|\sqrt{\beta_t}\theta - \min_{x\in\mathbb{R}^d}\alpha_{UCB}(x;\mathcal{D}_{t-1})| = 0$ can happen.

*# Derivation of Eq. (14)*

The two terms on the RHS of Eq. (13) are monotone increasing functions, and $\gamma$ is smaller than both numbers in the set,

hence each term in (13) is smaller than the value of the corresponding function operating on each number.

*# A) Derivation of Eq. (15). B) Did you consider the probability for $\max_{x\in\mathcal{D}_t}\alpha_{LCB} \leq \max_{x\in\mathcal{D}_t}f(x)$ to hold?*

A) There is a typo in (15), the 1st inequality should be $\max_{x\in\mathcal{S}_k}f(x) - \max_{x\in\mathcal{D}_t}f(x) \leq \mu_{t-1}(x_t) + \beta_t^{1/2}\sigma_{t-1}(x_t) +$

$1/t^2 - \max_{x\in\mathcal{D}_t}f(x)$. This follows Lemmas 5.7 and 5.8's proof in Srinivas et al [19]. For the 2nd inequality, with

the chosen $\beta_t$, following Lemma 5.5 in Srinivas et al [19], $\max_{x\in\mathcal{D}_t}\alpha_{LCB}(x;\mathcal{D}_{t-1}) \leq \max_{x\in\mathcal{D}_t}f(x)$. B) Yes. The

chosen $\beta_t$ ensures $\alpha_{LCB} \leq f(x) \leq \alpha_{UCB}$ ($x \in \mathcal{D}_t$) with probability $\geq 1 - \delta$ (Lemmas 5.5, 5.7 in Srinivas et al [19]).

*# The values of $a_k$ and $b_k$ in our experimental settings*

We set $a_k = 1$ and derive an expression for $b_k$ based on the kernel hyper-parameters and the search space size. The

exact formula is in lines 141-145 of our script BO_unknown_searchspace_good.py or other GP-UCB based scripts.

*# The relation of iteration $T$ and $\epsilon$*

Using our Lemma 5.1's proof, Lemma 5.8 and Theorem 2's proof in Srinivas et al [19], the regret bound $r_b(T) \leq$

$\sqrt{C_1\beta_T\gamma_T}/T + 2/T$ with probability $\geq 1 - \delta$. Thus, for any $T \geq (\sqrt{C_1\beta_T\gamma_T} + 2)/\epsilon$, the $\epsilon$-*accuracy* is satisfied in

each search space. Hence, we can see that when $\epsilon$ is smaller, $T$ needs to be larger to guarantee the $\epsilon$-*accuracy* condition.

————-REVIEWER 2————-

*# Clarity and organization must be improved according to detailed comments*

Yes, we will, using all the comments and suggestions from all the reviewers.

*# Instead of being an expansion of UCB, if it was general, it would have ranked higher*

GP-UCB is chosen as it has ability to analyze convergence, which is very important in the unknown search space setting.

Note that it is still possible to use GP-UCB to define the expanded search space, then other acquisition functions can

be subsequently used in this expanded search space to find optimal. However, in this case, the $\epsilon$-*accuracy* cannot be

guaranteed (e.g. EI convergence can be shown only in noiseless setting, PI/ES/PES do not have convergence proof yet).

*# The importance of our work*

Our major contributions are: 1) formalizing the convergence analysis of Bayesian optimization when search space

is unknown; and 2) proposing an effective algorithm that guarantees $\epsilon$-*accuracy* convergence. To the best of our

knowledge, there is no previous work that guarantees its solution converges to a point close to the objective function

global optimum when the search space is unknown. In fact, we can always find counter examples which show that with

high probability, the solutions of previous works do not converge to a point close the objective function global optimum.

*# Comparison with other NeurIPS submitted papers this year*

This comparison disadvantages us as we are not in a position to defend our method against something we do not have

access to. Thank you for the compliment though.

————-REVIEWER 3————-

*# More experiments with higher dimensional function (d=10)*

We have conducted more experiments with the 10-dimensional functions Ackley10 and Levy10. The setups are same as in the paper. For Ackley10, #experiments=30 whilst for Levy10, #experiments=10 as GPUCB-FBO computation time for Levy10 is so expensive that we can only get 10 experiments during the whole rebuttal time (GPUCB-FBO's average runtime is 239.83 seconds/iteration while our method average runtime is 21.15 seconds/iteration). For Ackley10, our proposed method outperforms other 6 methods by a high margin and is better than GPUCB-FBO,

and, note that GPUCB-FBO computation time is at least 5-6 times slower than our method. For Levy10, our proposed

method is slightly better than EIH, EI-vol2 whilst outperforming other baselines significantly.

[Meta-Review · NeurIPS 2019]

This paper proposes an algorithm to expand the search space for Bayesian optimization. The reviewers thought the work tackles an important problem and would be of interest to the community. The claims are well supported by empirical evidence and the paper is clearly written. There were concerns about the practicality of the method and that the work is a combination of well-known techniques. Because the paper presents a relatively novel approach and substantiates the claims with strong supporting evidence, it seems to be above the bar of acceptance.